# Self-Reported Dietary Choices and Oral Health Care Needs during COVID-19 Quarantine: A Cross-Sectional Study

**DOI:** 10.3390/nu14020313

**Published:** 2022-01-13

**Authors:** Elzbieta Paszynska, Szczepan Cofta, Amadeusz Hernik, Justyna Otulakowska-Skrzynska, Daria Springer, Magdalena Roszak, Aleksandra Sidor, Piotr Rzymski

**Affiliations:** 1Department of Integrated Dentistry, Poznan University of Medical Sciences (PUMS), 60-812 Poznan, Poland; amadeusz.hernik@gmail.com (A.H.); justynao@ump.edu.pl (J.O.-S.); 2Department of Pulmonology, Allergology and Respiratory Oncology, Poznan University of Medical Sciences (PUMS), Szamarzewskiego 82/84, 60-569 Poznan, Poland; scofta@ump.edu.pl (S.C.); daria.springer.pl@gmail.com (D.S.); 3Department of Computer Science and Statistics, Poznan University of Medical Sciences (PUMS), 60-806 Poznan, Poland; mmr@ump.edu.pl; 4Department of Environmental Medicine, Poznan University of Medical Sciences (PUMS), 60-806 Poznan, Poland; aleksandrasidor13@o2.pl (A.S.); rzymskipiotr@ump.edu.pl (P.R.); 5Integrated Science Association (ISA), Universal Scientific Education and Research Network (USERN), 60-806 Poznan, Poland

**Keywords:** COVID-19/SARS-CoV-2, oral epidemiology, oral health, public dentistry, dietary habits, BMI, overweight

## Abstract

The COVID-19 pandemic significantly impacted the healthcare system, including dentistry. However, it is not entirely clear whether affected patients’ willingness for regular dental visits and preventive behaviors with regards oral hygiene and diet. This is essential to understanding the potential effects of the COVID-19 pandemic on the acceleration of dental issues in the future. It was aimed at checking the level of dental visit avoidance, self-reported oral health needs, and dietary changes. This cross-sectional questionnaire study conducted in Poland (*n* = 2574; mean age 44.4 ± 15.6; female 56.3%) assessed nutritional habits and dental care changes during the COVID-19 pandemic. As demonstrated, nearly half of the responders (47.1%) avoided regular dental visits, while only 0.5% used online consultations. Fears related to potential cross-contamination in dental offices dropped from 25% to 11.4% and were associated with increased BMI and age (*p* < 0.05). Sweet snacking/drinking confirmed 19.1%/33.2% subjects. Self-reported oral health care needs (tooth stain, calculus, gingivitis, loss of fillings) were related to frequent snacking and poor oral hygiene (*p* < 0.05). The study highlights that pandemic periods are covered by eating and drinking changes combined with inadequate hygiene and dental care impose health complaints in the oral cavity. This can magnify both nutritional and interrelated oral health issues, highlighting the need to implement preventive and mitigation measures.

## 1. Introduction

The COVID-19 pandemic had a significant impact on the functioning of the healthcare system, including dental services. As the main route of SARS-CoV-2 transmission is through airborne droplets, dental staff were required to use personal protective equipment such as suits, goggles, face visors, and face masks. This decreased the risk of viral spread through dental bioaerosol generated when using high-speed handpiece, ultrasonic devices, air-water syringe, air-abrasion equipment, all together remaining suspended in the air for a long time, increasing the risk of transmission [1]. To avoid in-office transmissions, the number of patients allowed in the waiting room was limited. At the same time, the pandemic, temporary lockdowns, and fear of contracting the infection generally increased public anxiety due to the novel epidemiological threat [2]. This could affect patients’ behaviors in the context of willingness for regular dental visits and preventive measures related to oral hygiene and diet. 

However, it is known that the COVID-19 pandemic significantly affected dietary habits to a different extent in various populations [3,4,5]. This was particularly evident in the case of lockdown periods. In Poland, increased overall food consumption and snacking were observed and resulted in weight gain in nearly 30% of surveyed [6]. The dietary habits of individuals with higher BMI had been most adversely affected during the lockdown, resulting in less frequent consumption of vegetables, fruits, and legumes, while higher adherence to meat, sweets, dairy, and fast foods [6].

Whether oral hygiene habits were affected and to what extent during the COVID-19 pandemic was not subject to many investigations. This cross-sectional study conducted among adult Poles aimed to assess whether the COVID-19 pandemic affected the dietary choices, oral hygiene habits, and willingness to visit the dental office. This is essential to understanding the potential effects the COVID-19 pandemic may have on the acceleration of dental issues in the future

## 2. Materials and Methods

### 2.1. Study Group

An anonymous, self-designed questionnaire was prepared to examine the effects of a nationwide nutritional quarantine and dental care needs (Appendix A). The survey was conducted among individuals vaccinated at the main COVID-19 vaccination point in Poznan (Poland) for two months, between March and May 2021. Participation in the study was voluntary. Individuals were optionally asked to fill in the questionnaire while signing regular documents before receiving a COVID-19 vaccine. During this time, the number of confirmed SARS-CoV-2 infections in Poland increased by 68%, from 1.71 million to 2.87 million, and 29,952 COVID-19-related deaths were noted. The number of individuals who received at least one COVID-19 vaccine dose increased from 2.19 million to 13.71 million (36% of the Polish population) [7].

The inclusion criteria were Polish nationality and age ≥18 years old. Specifically, the study aimed to assess:Level of SARS-CoV-2 fears during dental treatment and through contact with the dental office.Previous and present self-reported dental complaints and attendance characteristics in dental practices.A daily number of consumed meals, drinks, and snacks during quarantine.Alcohol consumption and smoking frequency in the general population.

Since the study was conducted during the challenging time of the pandemic, the food frequency assessment was simplified to avoid the negative effect of the length of the questionnaire on the response rate [8,9,10]. The demographic data on each surveyed individual included age, sex, level of education (primary, secondary, tertiary, or vocational), BMI (calculated from reported weight and height), history of SARS-CoV-2 infection, and work type during the pandemic. 

The questionnaire was pre-tested by the researchers, including the professional dentist and dietician. Given the size of the target population (defined as a group of unvaccinated adult Poles at the time of the study), it was calculated using Cochran’s formula [11] that at least 2401 eligible individuals should be surveyed to reach a margin level of 2% at the confidence level of 95%.

### 2.2. Ethical Approval

This cross-sectional research methodology assumed non-invasive methods of analysis, such as and questionnaire provided to the subjects taking part in the project. All subjects participating in the project were informed of the aims and type of research (anonymous questionnaire security) and agreed to attend the survey. Ethical review and approval were waived for this study. At the same time, written consent was not necessary due to the anonymous study design not meeting the requirements of a medical experiment according to Polish law. The study was conducted in accordance with the Declaration of Helsinki.

### 2.3. Statistical Analysis

The data were statistically elaborated with Statistica v. 13.1 (StatSoft Inc., Tulsa, OK, USA). Since the variables expressed in the interval scale (age and BMI) did not meet the assumption of Gaussian distribution (Shapiro–Wilk test; *p* < 0.05), non-parametric methods were used for statistical elaboration. To analyze the differences in age and BMI between two or more independent groups, the Mann–Whitney U test or Kruskal–Wallis analysis of variance (ANOVA) with Dunn’s post-hoc method were used, respectively. The categorical frequencies were transformed into a 7-point scale to assess whether consumption of each food product varied between groups (0—no changes, 1—extremely very low, 2—very low, 3—low, 4—medium, 5—high, 6—very high 7—extremely very high). The correlations between age and BMI and weight change were evaluated with Spearman’s correlation coefficient. A value of *p* < 0.05 was considered statistically significant.

## 3. Results

### 3.1. Demographic Characteristics

A total of 2574 participants took part in the survey by filling in the questionnaires. The surveyed individuals were 18–93 years old, inhabited urban areas, both genders (*p* > 0.05). The demographic breakdown of the studied population is presented in Table 1.

Overall, 18.6% of respondents were infected with the SARS-CoV-2 virus. Age variable comparison evidenced a significant difference between age and pandemic morbidity that younger persons were more frequently affected from COVID-19 (mean age 41.7 ± 14.5) than older subjects (mean age 43.9 ± 15.7), *p* = 0.008. 

Overall, 28.8% of surveyed subjects were quarantined. As many as 55.6% worked remotely and stayed at home due to the pandemic. Home office work was more frequent in younger subjects (mean 38.2 ± 14.1 y.o., *p* < 0.001) with normal BMI (mean 24.4 ± 4.8 kg/m^2^, *p* < 0.001. Pearson Chi-square test showed that women (57.2%) more often worked remotely than males (*p* = 0.03).

### 3.2. Eating and Drinking Habits

Overall, 13.4% (*n* = 345) of surveyed declared increasing eating frequency compared to pandemic time. Interestingly, almost 1/5 subjects as 19.1% (*n* = 492) confirmed sweet snack preferences. Related to the period of the COVID-19 pandemic, the last meal was usually consumed from 6 p.m. 25.2% (*n* = 639) to midnight 4.3% (*n* = 110) with preferable 8 p.m. o’clock 51.5% (*n* = 1303). During the pandemic, almost a quarter of surveyed subjects changed their drinking habits 23.5% (*n* = 605); there was high variability in the type of drink that increased in the pandemic (Figure 1). One-third (33.2%, *n* = 855) of responders declared to sweeten warm drinks with sugar.

### 3.3. Smoking and Alcohol Consumption

The majority of those surveyed, 80.5% (*n* = 2072), did not report any tobacco smoking or another form of tobacco use during the COVID-19 pandemic. The number of smokers less than a pack a day was 16% (*n* = 412) and more than one pack a day 3.5% (*n* = 90). In contrast, only a minority, 17.3% (*n* = 448), did not report any alcohol consumption, half of the subjects declared drinking alcohol occasionally 53.1% (*n* = 1367), once a week 16.4% (*n* = 422) and potentially pleaded guilty to alcohol misuse 1.9% (*n* = 49) persons, who reported drinking alcohol every day during the COVID-19 pandemic. 

### 3.4. Attendance to Dental Care during the Pandemic

Only half of the respondents (52.9%) had a dental visit in 2020; significantly younger individuals characterized by normal weight/BMI visited dental practices (*p* < 0.001 in all cases). In the total group, 17.9% (*n* = 461) participants reported difficulties arranging an appointment in dental clinics. There were also 19.6% of patients (*n* = 506) who postponed a scheduled dental visit. Significantly older subjects often delayed dental appointments (*p* < 0.001). Generally, only 0.5% (*n* = 14) subjects used teledentistry for a virtual consultation. On-time of completing the questionnaire respondents reported the following self oral care needs as mineralized dental plaque deposit and tooth discoloration 20.5% (*n* = 528), tooth hypersensitivity 12.3% (*n* = 316), filling loss 8.3% (*n* = 213), gingival bleeding 7.3% (*n* = 189). In 2020, during the first period of pandemic time, 25% (*n* = 643) of surveyed participants were afraid to schedule a dental examination or any invasive dental treatment. In 2021, the number of persons with dental fear to be treated in dental practices decreased up to 11.8% (*n* = 304). Individual fears to be dentally treated were more frequent among older persons with higher BMI (respectively, for age *p* < 0.001; for BMI *p* = 0.04).

Questionnaire significant results according to age and BMI of subjects are presented in Figure 2 and Figure 3.

### 3.5. Correlations

Significant associations were found between the subjects’ age and BMI with selected parameters such as the number of dental visits, increased number of daily meals, increased number of sweet snacks, evening meals, and tobacco smoking (Table 2).

## 4. Discussion

Due to the COVID-19 pandemic in Poland, the Ministry of Health published on 26 March 2020, updated recommendations on how to proceed with the provision of dental services. It was recommended to verify the reason for reporting to the dentist by phone, allowing access only to patients who require urgent medical attention. As a result, the scope of dental services provided in the office as part of general primary care was limited to performing procedures such as severe pain, inflammatory and purulent processes, injuries, cysts, and conditions with a high risk of complications in patients. It was recommended to postpone all preventive/checkups, orthodontic, prosthetic, and periodontal visits. Patients suffering from COVID-19 or those quarantined requiring urgent care were referred only to special units under high aseptic restrictions. 

When comparing the ratio of patients/dental visits, it seems that pandemic regulations together with pandemic waves should be taken into account. Despite WHO guidelines, each country introduced different restrictions and bans. Waves of the disease have occurred throughout the world at certain intervals, and in some poor regions, it has not been possible to control and record infections. The media, government policy, and enforcement restrictions also played a key role.

Based on the published data from 2020 reports concerning patients’ behaviors, the pandemic has negatively affected dental attendance; as demonstrated by Peloso et al. [12], 44.2% of examined Brazilian patients responded that only in case of dental emergency would come for the appointment, and 17.5% rather avoid dental visits. Nair et al. [13] showed that almost 18% of surveyed subjects experienced oral emergencies under lockdown and quarantine. Sekundo et al. [14], based on telephone interviews, estimated that as many as 16.2% applied for dental treatment due to toothache, dentine hypersensitivity, filling loss, gingivitis/periodontitis.

In another study conducted in Madrid by Gonzalez-Olmo et al. [15,16], every fourth examined subject would avoid dental attendance to dental treatment due to the reasons mentioned above. It was calculated that among individuals older than 60 y.o., as many as eight times more anxiety levels occurred.

Surveys conducted in Poland revealed that dental anxiety levels increased during the pandemic compared to the pre-pandemic period [17]. In another research conducted in Poland, one-fifth of surveyed patients reported a significant fear connected to dental treatment during the COVID-19 pandemic [18]. Our study showed that every fourth patient was afraid to visit the dentist’s office under pandemic time. However, in present time, when the questionnaires were collected, the level of stress related to dental treatment subsided declined. Perhaps reducing anxiety is likely due to the high vaccination level of medical personnel in Poland [19]. It is possible that these patients appreciated the efforts of the dental staff regarding the high standard of decontamination of the operating field and office space. Unfortunately, operative dental care is connected with an increased risk of infection mainly through bioaerosol generated during intraoral procedures performed with water spray and later spreading in the air. Therefore, dental treatment is linked to the risk of acute respiratory infections [20]. In the case of COVID-19, all patients should be considered as possible carriers of SARS-CoV-2 due to mild symptoms or asymptomatic patients and bioaerosol issues [21]. This potential possibility of cross-contamination puts dentists at a high risk of infection transmission and forces them to follow the rules in rapidly changing pandemic times [22]. Recommendations for dental teams regarding dental visits during a pandemic additionally include the use of mouth rinses based on H_2_O_2_ or chlorhexidine (CHX) before examination of the oral cavity, isolation of the treatment area with a rubber dam; four-handed work; and the use of robust suction systems [22,23]. In the present study, the percentage of persons still afraid of visiting the dental office fell by half. Perhaps their decision was depended on the vaccination procedure and amount of more and more vaccinated population groups.

The present study showed that a minority of surveyed (0.5%) subjects used teledentistry for a virtual consultation. The new experiences of the COVID-19 pandemic show how important a role in doctor-patient communication is played by the development of information and communication technologies and the ability to remotely solve dental problems that do not require a visit to a dentist’s office [24,25]. It is estimated that about two-thirds of the Polish population is willing to use additional medical services online, and about 40% of patients actively search for medical services on the internet [26]. Teledentistry may support in COVID-19 oral care in case of non-invasive interventions [27]. Unfortunately, state national insurance did not decide to reimburse dental teleconsultations, only dental e-health service is available in the private sector [28].

Based on information from the literature, clinical situations that do not necessarily qualify as urgent dental interventions include lack of temporary fillings in deciduous and permanent teeth, not accompanied by pain; mucosal injuries; pericoronitis around permanent molars, inflammation of the gingival pocket [29,30]. There are more clinical indications in pediatric dentistry that do not require an immediate dental visit, such as the oral mucosa condition during tooth eruption, exfoliation of primary teeth, and preventive measures together with dental education advice can be continued by caregivers [31].

The present study results support the opinion from another survey that pandemic time may alter dietary habits in the population [6]. Generally, overeating may be even more severe during quarantine due to extended stays at home with often unlimited access to food [32]. As shown in the present study, the increase in snacking may lead to a rise in the tendency to eat meals in the evening, which harms the oral cavity more than snacking during the day due to the inhibition of salivary secretion in the night rest [33]. This is significant mainly for population groups affected by hyposalivation, then typical dental complaints followed by an absence of salivary clearance may occur [34]. Almost a quarter of surveyed subjects changed their drinking habits and preferred sweetened snacks and drinks during the pandemic. It appears that sugar consumption was maintained on a relatively high level, and changes in dietary patterns may potentially impulse weight rise and stimulate unlimited dental caries [31]. A similar marked increase in snacking and sweet consumption among Poles by Blaszczyk-Bębenek et al. [35] and Lithuanians by Kriaucioniene et al. [36] were observed. All observations in our survey indicate that nutritional awareness among Poles is insufficient and obliges more attention to promoting healthy food patterns. One example of lifestyle intervention regarding nutrition, physical activity and psychological support might be an Italian project “#StayHomeStayFit” posted online by University of Milan [37].

The present study shows that the subjective ailments were connected to a lack of proper oral hygiene and diet maintenance. Complaints on tooth discoloration, dental calculus, or halitosis were probably caused by dental plaque deposits, changes in oral hygiene habits, and limited contact with the dentist. It could be speculated that due to a lack of regular dental appointments, motivation to perform oral hygiene procedures lasted at a low level over a long period. Other studies showed similar findings and experiences of only urgent dental care in several countries under pandemic restrictions [11,38,39,40,41,42]. They described identical reasons: the delayed dental care due to private practice closures, lack of professional staff and personal protective equipment [38,39,40]. In the geographical area of our survey, during the initial period of the pandemic, 70% of dental clinics periodically suspended their activities [41,42]. All scheduled dental procedures that were not classified as an emergency were, in accordance with WHO recommendations, postponed [41]. Delayed dental treatment, but also poor prevention in the long-term effects of a pandemic may prove particularly acute for patients at high risk of oral diseases. Hence, the first signals of patients neglect we could observe in their list of ailments reported from our survey. It seems necessary to develop new standards for dealing, building new ways of communicating with patients from all age groups, but also to create programs adapting to the new circumstances.

The present study provides an overview of dental care attendance and dietary behaviors during quarantine, but its results cannot be interpreted in the context of long-term effects—this would require further follow-up investigations. It should also be stressed that any oral or general health statuses data were not obtained by medical or dental examinations. The research was based on a survey that excludes the possibility of verifying the data on objective grounds. The survey was conducted at the vaccination point, which allowed to obtain a relatively high number of questionnaires but may limit access to some societal groups. We also were not able to extend questions about particularly diet/nutrition before and after pandemic due to limited frame of the questionnaire collected at the vaccination point.

## 5. Conclusions

The present study uncovers various directions in which dietary habits and oral health may be altered during the COVID-19 pandemic. It highlights that oral hygiene was neglected, while food consumption generally increased. This may be due to several reasons, including anxiety, stress, fear of dental visits, and overall decreased social mobility. Due to diminished access to dental practitioners, a significant need to support appeared, particularly for sensitive groups such as overweight and older-aged subjects, and those not motivated or trained to maintain proper oral hygiene levels and diet during future epidemic-related quarantines. The results of this study highlight the role that should be played in this regard by groups of nutritionists, dental professionals, and public health practitioners.

## Figures and Tables

**Figure 1 nutrients-14-00313-f001:**
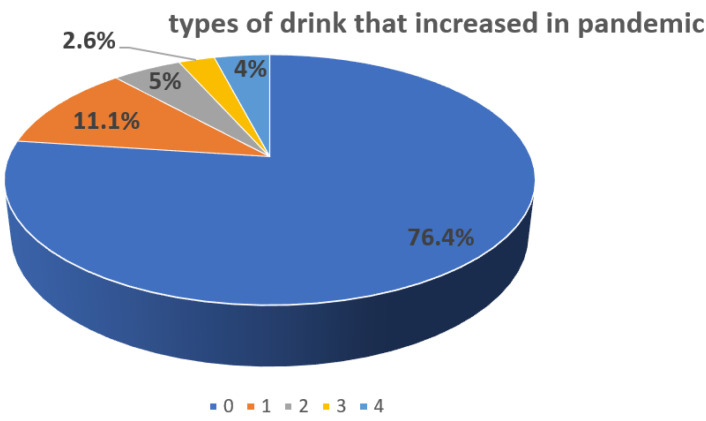
Variability in the type of drink that increased in pandemic divided into categories 0—no changes (76.4%), 1—water (11.1%), 2—coffee and tea (5%), 3—carbonated acid beverages (2.6%), 4—alcohol beverages (4%).

**Figure 2 nutrients-14-00313-f002:**
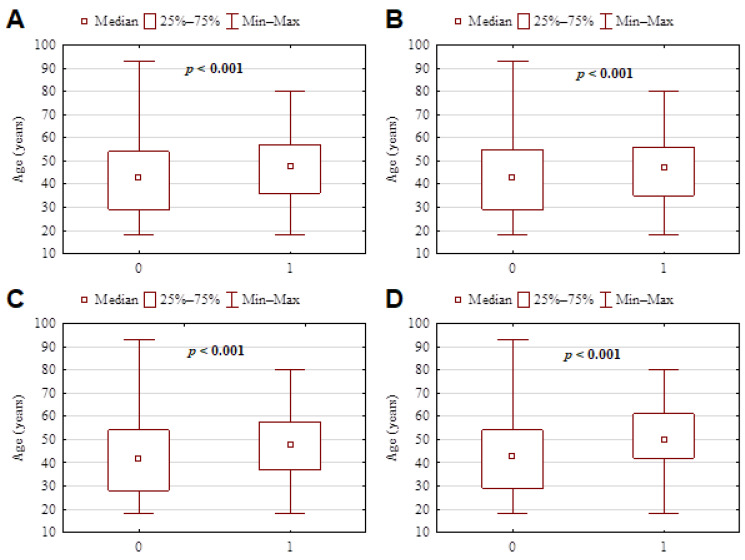
Questionnaire significant results according to the age of subjects. (**A**) Postponing a dental visit and the age of patients indicated that older subjects frequently postponed the appointment. 0—no, 1—yes (*p* < 0.001). (**B**) Difficulty in scheduling a dental visit was significantly greater among older persons. 0—no, 1—yes (*p* < 0.001). (**C**) In the first period of pandemic time, dental anxiety and fear to schedule a dental examination or any invasive dental treatment were significantly greater among the older population. 0—no, 1—yes (*p* < 0.001). (**D**) At the time of the survey, dental anxiety and fear to schedule a dental examination or any invasive dental treatment were still significantly greater among the older population. 0—no, 1—yes (*p* < 0.001).

**Figure 3 nutrients-14-00313-f003:**
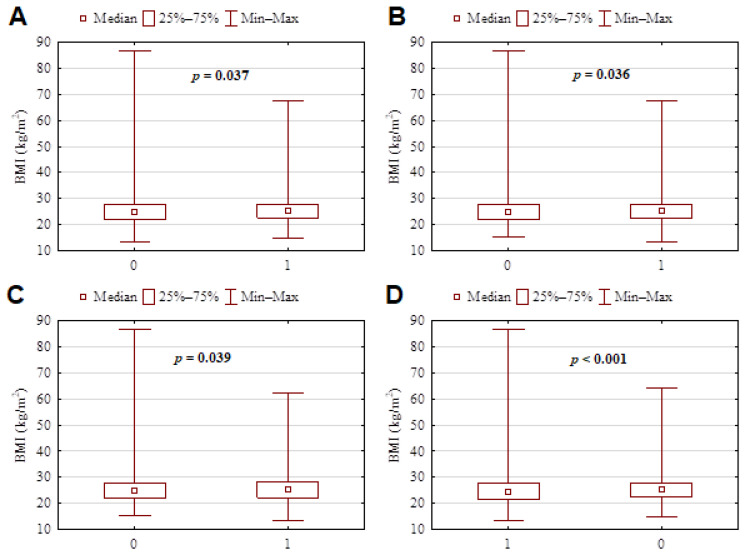
Questionnaire significant results according to BMI of subjects. (**A**) Difficulty in scheduling a dental visit was significantly greater among subjects affected from overweight. 0—no, 1—yes (*p* = 0.037). (**B**) In the first period of pandemic time to schedule for a dental examination or any invasive dental treatment, dental anxiety and fear were significantly greater among people characterized by high BMI. 0—no, 1—yes (*p* = 0.036). (**C**) At the time of the survey, dental anxiety and fear to schedule a dental examination or any invasive dental treatment were still significantly greater among the overweight population (high BMI). 0—no, 1—yes (*p* = 0.039). (**D**) Attendance to dental care during the pandemic was confirmed by subjects with normal BMI 0—no, 1—yes, (*p* < 0.001).

**Table 1 nutrients-14-00313-t001:** Demographic breakdown of surveyed participants (*n* = 2574).

**Age** (years) mean ± SD (min–max)	44.4 ± 15.6 (18–93)
18–25, *n* (%)	451 (17.8)
26–35, *n* (%)	380 (15.0)
36–45, *n* (%)	455 (18.0)
>45 *n* (%)	1248 (49.2)
**Gender** Female, *n* (%)/Male, *n* (%)	1429 (56.3)/1107(43.7)
**Weight** (kg) mean ± SD (min–max)	74.8 ± 17.0 (34–190)
**BMI** (kg/m^2^) mean ± SD (min–max)	25.3 ± 4.9 (13.6–86.7)
Underweight (<18.5), *n* (%)	87 (3.4)
Normal weight (18.5–24.9), *n* (%)	1207 (47.8)
Overweight (25–29.9), *n* (%)	903 (35.8)
Obesity (≥30), *n* (%)	329 (13.0)
**Education**	
Primary *n* (%)	51 (2.0)
Vocational *n* (%)	269 (10.5)
Secondary *n* (%)	902 (35.3)
Higher *n* (%)	1332 (52.2)

BMI—Body Mass Index; SD—standard deviation.

**Table 2 nutrients-14-00313-t002:** The Spearman’s correlation rank tests showed significant results regarding age, BMI, and selected parameters (*p* < 0.05) for all surveyed participants.

Group *n* = 2574	*p*-Value	Spearman R
age and dental visits	<0.001	−0.09
age and increased meals	0.017	0.05
age and increased sweet snacks	<0.001	−0.07
age and evening time meal	<0.001	−0.29
age and tobacco smoking	0.008	0.05
BMI and dental visits	0.004	−0.06
BMI and increased meals	<0.001	0.08
BMI and evening time meal	<0.001	−0.10
BMI and increased tobacco smoking	<0.001	0.07

## Data Availability

Data associated with the paper are not publicly available but are available from the corresponding author at reasonable request.

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
