# Peer review of "Self-Reported Dietary Choices and Oral Health Care Needs during COVID-19 Quarantine: A Cross-Sectional Study"

_nutrients, 2022, doi:10.3390/nu14020313_

Round 1

Reviewer 1 Report

The Authors must see my remarks

Author Response

Dear Reviewer #1

Please se the attachment (uploaded)

Reviewer 2 Report

Dear authors, thank you for submitting the manuscript to Nutrients journal.

  1. The title should be focused on nutrition, according to the aim and the scope of the journal, not on oral health need needs. Please rephrase it.
  2. How was the survey validated? It should be validated by experts.
  3. The same observation for the abstract. Please rephrase: ”It was aimed at checking the level of dental visits avoidance, self-reported oral health needs, and dietary changes”. Also, reformulate the last sentence.
  4. Please explain ”geriatric dentistry” as a keyword. Similar, for page 2, line 59-60: ”This cross-sectional study conducted among adult Poles aimed to assess whether the COVID-19 pandemic affected the visits at the dental office, oral hygiene habits, and diet”.
  5. Page 2, line 51-52: please add a reference. The same suggestion for and line 52-53. Similar, for line 54: ‘‘In Poland, increased overall food consumption and snacking 53 was observed and resulted in weight gain in nearly 30% of surveyed”.
  6. Page 2, line 76-77. Please explain: ”Specifically, the employed questionnaire was relevant from the onset??? of the Covid-19 pandemic to the questionnaire collection time (from March to May 2021)”.
  7. Page 2, line 83. ”work type” - where does it come from in the questionnaire?
  8. Related to the questionanire in Supplementary material: please revise: Question 3 it in not translated in English. See Q 5: releadvice?; Q 7: check-up, please modify. Q 10: ”any simptoms?”Q 26: please change: ”mpst frequently prefered mouthwash?”; Q 30: ”Does the prosthesis cause discomfort / discomfort?”. Related to Q 35, please explain the categories found in the manuscript, they are different (see ” level of education (primary, secondary, tertiary, or vocational” – line 89).
  9. Page 2, line 86-88: I understand the reasoning related to the length of the questionnaire, but related to the topic of the article, how do you justify it? It could be reduced to other categories, such as the prosthetic part. Please explain.
  10. Pleas explain: ”Dichotomous questions were applied to 90 identify current smokers and those suffering from alcohol addiction”. Why is it relevant to the study? The only mention in the text is: ”The number of alcohol addicts in the study who declared each day drinking was ra-286 ther low (n=49)”. Also, change ”ther”!.
  11. Ethical approval: Given the affiliation of the authors (also mentioned in the introduction of the questionnaire), the agreement of the ethics committee of the university was required. Please explain why the authors did not get it! (I do not think that Polish law excludes the need for ethical agreement). Also, please consider the level of the journal you aimed to publish the manuscript! At least the consent of the vaccination center manager should have been obtained. Please mention the Helsinki Declaration. The purpose of the study was not specified to the respondents!
  12. Page 2, line 108: ”the categorical frequencies were transformed into a 7-point scale”. Please explain.
  13. Page 2, line 112: Demographic Characteristics: why did you used upper case for C?
  14. Page 2, line 114: what is the purpose of the word ”Generally”?
  15. Page 2, line 115 and 118: what is the purpose of using and (p>0.05) and Mann–Whitney U test, t-test, and Welch test for Table 1 (descriptive statistics)?
  16. Page 2 and page 3, lines 124-128 : ”as many....” Why is it relevant to the study?
  17. Page 4, line 132-133: ”The last meal was usually consumed from 6 p.m. 25.2% (n=639) to midnight as 4.3% (n=110) with preferable 8 p.m. o’clock as 51.5% (n=1303)” Is the question general or pandemic?
  18. Page 4, line 141: Smoking and alcohol consumption are general or pandemic?
  19. Page 7, line 206. ”There are few reports concerning patients behaviors under pandemic waves from 206 2020”. Really? Please add some references.
  20. Page 7, line 221: Please explain: ”Our study showed that at the beginning of the pandemic, every fourth 221 patient was afraid to visit the dentist's office”. The authors also mentioned: ”during the third wave of Covid-19 disease”…..
  21. Page 7, line 227: Add a reference for ”Population Perhaps reducing anxiety is likely due to the high 227 vaccination level of medical personnel in Poland, over 90%”. Has the population been informed about this?
  22. Page 8, line 282-283: ” Moreover, a fast-food diet and high carbo-282 hydrate consumption has been associated with pro-inflammatory effects‘‘. It is not clear, so please explain.
  23. For the Results: please emphasize the results related to diet/nutrition, before and after pandemic (see questionnaire, items 12, 13 and 14). In fact, there are the only questions related to dietary choices/changes. Please add the results, comparatively. It is mandatory! Of course, revise the main text, the results part) and the abstract, accordingly.
  24. Page 9, line 319: ” The survey highlights that pandemic lockdown periods affect eating and 319 drinking habits in assistance of dental ailments caused by hygienic neglect in oral cavity” Also caused by stress, anxiety? Please rephrase.
  25. Page 9, line 323: ”diet regime”? Please reformulate.
  26. Please add some comments in the conclusions related to implications and role of dental practitioners, and also for public health practitioners.
  27. The results should be compared with similar studies conducted in similar countries in the discussion part. Please add some references.

Author Response

Dear Reviewer #2

Please see the attachment (uploaded)

Round 2

Reviewer 2 Report

I agree with the publication of the manuscript in the present form.

This manuscript is a resubmission of an earlier submission. The following is a list of the peer review reports and author responses from that submission.